# Blurring the Boundaries between a Branch and a Flower: Potential Developmental Venues in CACTACEAE

**DOI:** 10.3390/plants10061134

**Published:** 2021-06-03

**Authors:** Isaura Rosas-Reinhold, Alma Piñeyro-Nelson, Ulises Rosas, Salvador Arias

**Affiliations:** 1Instituto de Biología, Jardín Botánico, Universidad Nacional Autónoma de México, Ciudad de México C.P.04510, Mexico; isaurarosas@ciencias.unam.mx (I.R.-R.); urosas@ib.unam.mx (U.R.); 2Posgrado en Ciencias Biológicas, Instituto de Biología, Universidad Nacional Autónoma de México, A. P. 70-153, Ciudad de México C.P.04510, Mexico; 3Departamento de Producción Agrícola y Animal, Universidad Autónoma Metropolitana-Xochimilco, Ciudad de México C.P.04510, Mexico; almapineyro@gmail.com; 4Centro de Ciencias de la Complejidad (C3), Universidad Nacional Autónoma de México, Ciudad de México C.P.04960, Mexico

**Keywords:** flower development, floral shoot, flower evolution, cacti evolution, evo-devo, flower organ identity

## Abstract

Flowers are defined as short shoots that carry reproductive organs. In Cactaceae, this term acquires another meaning, since the flower is interpreted as a branch with a perianth at the tip, with all reproductive organs embedded within the branch, thus giving way to a structure that has been called a “flower shoot”. These organs have long attracted the attention of botanists and cactologists; however, the understanding of the morphogenetic processes during the development of these structures is far from clear. In this review, we present and discuss some classic flower concepts used to define floral structures in Cactaceae in the context of current advances in flower developmental genetics and evolution. Finally, we propose several hypotheses to explain the origin of these floral shoot structures in cacti, and we suggest future research approaches and methods that could be used to fill the gaps in our knowledge regarding the ontogenetic origin of the “flower” in the cactus family.

## 1. Introduction

Flowers, unlike other organs such as leaves, roots and stems, are composite structures made of a number of organs that form an ordered pattern [1]. The basic flower structure or floral ground plan is the result of key innovations in angiosperms that may have originated from coadaptations with pollinators [1,2]. Some of these key novelties are bilateral symmetry, the specialization of the perianth into sepals and petals [3] and the gynoecium composed of carpels [4]. Molecular clock dating studies have suggested that flowering plants originated in the late Triassic period, ~209 million years ago [5]. These studies also suggest that the core group of angiosperms appeared in the Jurassic, while the Cretaceous period observed the emergence of multiple other diversifications across flowering plants (~140–90 mya; [5]).

Most angiosperms have a conserved floral organization that consists of four whorls of concentric organs. This structure is commonly organized into two whorls of sterile organs, i.e., sepals and petals, arranged as the first and second whorls, respectively, and then a whorl of stamens and, in the innermost whorl, the carpel, which is a novel structure enclosing the ovules, not present in other seed plants [2]. Such an organization seems to be genetically determined by the interplay of a set of homeotic genes that interact in a whorl-specific manner. This was documented in floral mutants of *Arabidopsis thaliana* and *Antirrhinum majus*, which were used as model species in flower developmental genetics, giving way to the so-called ABC model [6,7]. This model posits that the development of the four whorls that typically comprise a flower is directed by the spatiotemporal expression and interaction of three gene classes. The expression of class A genes in the outermost section of the flower meristem (*APETALA1* (*AP1*) and *APETALA2* (*AP2*)) gives way to sepals; the concerted expression of class A and B genes (*APETALA3* (*AP3*) and *PISTILLATA* (*PI*)) determines petal formation; the concerted expression of class B and C genes (*AGAMOUS* (*AG*)) determines the development of stamens in the third whorl; and, finally, the expression of the class C gene alone originates the carpels in the fourth whorl [6]. Furthermore, all genes except for *AP2* are part of the MADS-box type II gene family of transcription factors [6]. An additional category of genes, *SEPALLATA* (*SEP*) or E class genes, characterized years later, is expressed across the floral meristem, and while four paralogs have been documented in *A. thaliana*, *SEP3* is part of the whorl-specific protein tetramers that, together with the ABC class genes, underlie the differentiation of each whorl [8,9]. Due to their functional and sequence conservation across angiosperms, ABC genes have been studied from an evolutionary and developmental perspective, testing whether variations in the spatiotemporal expression of orthologs due to subfunctionalizations, neofunctionalizations or new protein–protein associations could underlie diverse floral morphologies [10,11]. Thus, ABCE genes have become a useful model to test hypotheses regarding how development has evolved, leading to morphological variations in flowers. For example, molecular evolution as well as spatiotemporal patterns of expression in developing flower primordia in non-model plants such as species within the Ranunculaceae [12], Triuridaceae [10,13,14], Aizoaceae [15] or Orchidaceae [16] have been used to analyze both the functional conservation of ABC class gene activity and its variants across angiosperms, broadening our knowledge of the genetic bases of floral organ diversification and opening new avenues of research regarding the molecular underpinnings of perianth evolution [15,17].

In this paper, we review the current understanding of embryological, histological, and genetic data underlying flower development in Cactaceae, a family comprising approximately 1438 to 1870 species [18,19]. Some of these taxa are remarkable as crop species (i.e., *Opuntia ficus-indica*, *Selenicereus undatus*) or charismatic ornamental species; nonetheless, and despite their economic importance and botanical allure, the origins of their floral novelties and floral developmental patterns are far from fully understood. In this work, we will focus on two singular structures whose development has received little attention in this group of plants: (1) the pericarpel, a seemingly vegetative tissue that encloses the portion of the receptacle where the ovary and the stamens originate; and (2) the perianth, which shows a morphological gradation from bracts to petal-like structures in all cacti (see Figure 1).

Last, based on the current understanding of the molecular bases of reproductive structure induction and flower development in angiosperms, as well as our own observations in diverse cacti taxa, we propose several hypotheses that could explain the complex ontogenetic origin of the unique floral structures in this group, pointing to new research venues.

## 2. Evolutionary History and General Flower Structure in CACTACEAE

Cactaceae are a eudicot family within the Caryophyllales and have been organized into five subfamilies: Leuenbergerioideae, Pereskioideae, Maihuenioideae, Opuntioideae and Cactoideae [18,20,21]; see Figure 1 for a simplified phylogeny. The family diverged quite recently, at approximately 35 Mya, placing Cactaceae as a late arrival in angiosperm history. It is thought that the majority of species diversification occurred during the late Miocene, ≈10–5 Mya [22].

Cactaceae are distributed across a wide variety of ecosystems, from deserts to rainforests, and have a highly specialized vegetative axis where the majority of species seem to lack leaves as these are highly reduced, and develop succulent photosynthetic stems that, in turn, exhibit a wide variety of shapes (cylinder, barrel shape, flattened) and sizes [23]. For example, the diameter of the body in *Blossfeldia liliputana* is 10 mm and is considered the smallest plant in the family [24]. In contrast, *Echinocactus platyacanthus* can reach a width of 1.5 m.

In addition to the attractiveness of their stems, their astonishing flowers are another distinctive feature of this family, possessing a number of important characters with taxonomic importance, such as color, size, shape, presence or absence of spines, leaves and bracts [23]. Most attention on cactus flowers has focused on analyzing traits from these organs from a systematic perspective, yet the developmental events leading to the formation of these organs have been largely understudied [25,26,27,28,29,30,31,32,33,34,35]. This is the case for the evolutionary and developmental origins of the perianth and the pericarpel, two puzzling structures that require contemporary approximations to fill in the gaps pertaining to floral evolution in Cactaceae.

From a phylogenetic perspective, Cactaceae is a member of the Portulacineae clade, sister to Anacampserotaceae, and is closely related to Portulacaceae, Talinaceae, Basellaceae, Halophytaceae and Montiaceae [21,36,37,38]. A common theme among members of these families is that they have a meristematic ring from where stamens develop. In *Anacampseros*, the multiplication of stamens is divided into two groups: the first group comprises the multistaminated species characterized by the presence of a well-defined ring meristem, and the second group includes species with a low stamen number [39]. *Talinum*, *Portulaca* and *Calandrinia* are in the first group. Unsurprisingly, Cactaceae species also have a meristematic ring corresponding to the first group [25,26,27,29], equivalent to the one described in *Anacampseros* [39]. While the meristematic ring is a common feature in this group of plants, the genetic mechanisms underlying its formation remain to be fully investigated [40].

Despite their close phylogenetic relationship, the morphology of the “cactus flower” deviates from some features found in several members of the *Portulacineae*. For example, a widespread feature in families within this clade is that flowers are generally inserted within an involucre of two median bract-derived phyllomes, having a sepaloid appearance [41]. This is the case for species of *Talinum*, *Claytonia*, *Anacampseros* and *Calandrinia* [39,42,43,44], in which each flower is subtended by two involucral bracts that protect the young floral buds, covering the floral apex and the five tepals [39,42,43,44]. Regarding the nature of the involucre as bracts or sepals, the topic has been extensively discussed, although evidence for a sepal identity is weak and is contradicted by the morphology of the petaloid organs that resemble true sepals [41]. The presence of two bracts associated with the flower as well as petaloid tepals appears to be a synapomorphy for the clade including Portulacaceae and Cactaceae [41,45]; nevertheless, in the latter, involucral bracts are absent [41]. Furthermore, phylogenetic reconstructions of perianth characteristics in Caryophyllales show that in Portulacineae, the perianth is biseriate with bract- and sepal-derived organs; in Cactaceae, the perianth is multiseriate and has been argued to be composed of bract- and sepal-derived organs [41]. It is worth mentioning that this type of perianth is unique to the Cactaceae [41]. In short, some of the evolutionary novelties found in Portulacineae flowers are not shared in Cactaceae.

Flowers in cacti are usually sessile (without a stalk), with exceptions in *Pereskia* and *Leuenbergeria* (Figure 2a). They come in a variety of sizes: *Selenicereus* has the largest flowers (35+ cm), while species of *Epithelantha* have very small flowers (6 to 18 mm) [23,46,47]. Most flowers are bisexual [23], but some dioecious species have been reported, such as *Opuntia stenopetala* [30,48], *Opuntia robusta* [34] and *Mammillaria dioica* [32]. Cacti flowers are usually described as solitary [23], but a few species from *Pereskia* develop racemose inflorescences [49]. Furthermore, in *Myrtillocactus geometrizans*, a widespread columnar cactus from arid Mexican landscapes, it is common to find several flowers in clusters originating from a single vegetative areole [23,50,51].

Although cacti flowers are radially symmetrical, a few species have been described as asymmetrical [50], such as *Aporocactus flagelliformis* and *Zygocactus* species. This is supported by observations of the orientation of stamens and pistils, bending of the floral tube, the presence of an oblique tube throat or even the shape of the ovary, as reported in *Selenicereus spinulosus* [50]. Whether cacti species display true flower asymmetry remains to be further analyzed; however, the aforementioned observations are consistent with our studies in *Disocactus speciosus* and *D. kimnachii*, where preliminary data show that stamens are more abundant on the ventral side of the whorl rather than arranged as a homogeneous ring, and the same can be observed in the South American cacti genera *Trichocereus* and *Harrisia*. In contrast, in *Aporocactus flagelliformis,* there is a more prominent presence of stamens on the dorsal side of the whorl, while in its sister species, *A. martianus*, stamens have a ring disposition. Interestingly, variability between species in the same genus has also been observed in some *Selenicereus* species; for example, whereas stamens emerge as a ring in *Selenicereus undatus*, in *Selenicereus validus*, more stamens are present on the dorsal side. The abortion or prevalence of male organs on one side of the flower is part of the flower asymmetry syndrome documented in other angiosperms, such as the Lamiales [52]. These observations suggest that incipient flower asymmetry might be independently evolving in different clades of epiphytic cacti, but the underlying ontogenetic mechanisms have yet to be validated through developmental and genetic analyses.

The subfamilies Leuenbergerioideae and Pereskioideae (Figure 1, green and red clades, respectively) are the least “cacti-looking” lineages and are considered to contain the earliest diverging group in Cactaceae [49,53,54,55,56]. Species in the genera *Pereskia* and *Leuenbergeria* possess broad leaves, internodes and a minimum of succulence. The spines associated with axillary buds (the areole) are the diagnostic feature that groups them together with the rest of the Cactaceae [57].

Another relevant feature in *Pereskia* and *Leuenbergeria* is that true inflorescences are produced only in these genera: both paniculate and cymose inflorescences [23,49]. Another salient feature is that some species, such as *P. aculeata*, *P. diaz-romeroana* and *L. lychnidiflora*, have a superior to half-inferior ovary [26,27,28,31]. Moreover, the ovary in *L. lychnidiflora* is also multilocular, with several separate carpels, resembling axile placentation with pocket-like locules [26] (Figure 2a). The superior position of the ovary is considered to be an ancestral feature in the family [23], while the inferior receptacular ovary in the other subfamilies is derived (Maihuenioideae, Opuntioideae and Cactoideae; Figure 2b and c). According to developmental studies in *Pereskia* and *Leuenbergeria*, the ovary is in a transition process; thus, the definition of the ovary position in these genera is complex [31,58]. Nevertheless, a short tube showing an early tendency to form a receptacle can be observed in *P. aculeata* [50], with areoles and leafy bracts emerging from it.

Flowers in the subfamily Opuntioideae (the prickly pear group) (Figure 1, yellow clade) are structurally more complex than those in Pereskioideae and Leuenbergerioideae because the ovary is embedded within the axial tissue denominated the pericarpel [50,59] (Figure 2b). In *Opuntia*, the pericarpel bears stomata and areoles (axillary buds) on its surface [59]. In addition, the presence of mucilaginous cells, druses and parenchyma cells resembles those present in the stem [60,61]. Based on this evidence, it has been suggested that the pericarpel has an axial origin; in other words, the pericarpel is a vegetative tissue originating from a vegetative meristem within the areole. Due to the axial–appendicular nature of the tissue, Mauseth [60,61] and most contemporary cacti specialists refer to this unique and remarkable structure as a “flower shoot”.

These flower shoots are generally described as solitary, although in some exceptions, new flower shoots emerge from the areoles present on the pericarpel, forming a chain-like structure of flower shoots, a phenomenon that often occurs in *Cylindropuntia* species [60,62]. The axillary buds (areoles) on the flower shoots of *Cylindropuntia fulgida* (“chain fruit cholla”), *C. leptocaulis* and a few other species produce floral shoots that later become “fruits”, whose axillary buds repeat the process [60,61]. Such arrangement of branch flowers growing over one another could be considered an inflorescence-like formation pattern, but this is an issue that has to be more thoroughly analyzed. An example of solitary flower shoots is those present in prickly pears (*Opuntia* spp.), which are born at the apex of flattened stems. Their perianth segments are generally yellow to red (due to betaxanthin pigmentation), and the pericarpel often presents a round shape, with spines and deciduous leaves [56,59].

Maihuenioideae is a monogeneric subfamily from the Patagonian Desert [23,63]. *Maihuenia* comprises two species: *M. patagonica* and *M. poeppigii* [23]. This genus is characterized by its small-leaved and dense, compact, mound-forming plants [63]. *Maihuenia* was initially included inside Opuntioideae because it shares vegetative and floral features with the latter, as *Maihuenia* flowers are diurnal, white and yellowish in color, very similar to *Opuntia* flowers [63]. More recent phylogenetic analyses showed that *Maihuenia* is an independent group from the Opuntioideae subfamily, giving rise to a new subfamily: Maihuenioideae [64]. Nevertheless, given the limited information on the morphology and anatomy of flower shoots from Maihuenioideae, we did not include them in Figure 1.

The general idea of a cactus comprises species such as peyote (*Lophophora williamsii*) or the saguaro cactus (*Carnegiea gigantea*), both belonging to the Cactoideae. This subfamily displays the largest variation observed in flower shoots. Thus, flower shoots within Cactoideae species can have podaria, ribs, tubercles that cause cortical succulence, bracts with a reduced laminar part, elongated receptacle tubes, thickness or rigidity in the pericarpel [50,56]. Flower shoots in some columnar cacti, such as *Cephalocereus* and *Pachycereus*, are large in size and have nocturnal anthesis, bearing a white perianth and, in some cases, an unpleasant odor [65,66]. Large flower shoots can also be observed in the epiphytic species *Selenicereus undatus* (dragon fruit or pitahaya), which has one of the largest flowers in the family, reaching up to 29 cm in size [54]. In contrast, small flower shoots can be observed in different genera, such as *Mammillaria*, *Coryphantha*, *Lepismium* and *Rhipsalis*. These small flower shoots display a reduction in pericarpel tissue [50,56] as well as a reduction in or loss of axial characteristics of the flower shoot, such as the loss of areoles, scaly bracts and green cortex, in addition to the nonshowy color of the tube, as well as the transformation of bracts into petaloid structures [50] (Figure 2c).

## 3. Cactus Flower Development

According to Boke [57], the development of the so-called flower shoot in Cactaceae is similar to that in many other plants, and the most significant differences are observed in pistil and, in particular, carpel development. For this reason, most investigations on flower shoots have focused on the reproductive organs (androecium and gynoecium, [26,27,28,29]), their embryology [31] and the mechanisms underlying dioecy, such as those documented in *Opuntia stenopetala* [30,48], in *O. robusta* [34] and in *Mammillaria dioica* [32]. These studies have shown that dioecy in cacti is the product of programmed cell death and other developmental mechanisms, such as ovule abortion, rather than the lack of floral whorl determination.

In Cactaceae, the androecium has numerous stamens (multistaminated), which initiate in a uniform and prominent ring primordium, on which the stamens arise in a centrifugal succession [25,27,50,67]. This stamen ring is also observed in *Talinum* [43] and *Anacampseros* [39]. In the Caryophyllales, the multistaminate androecium presents morphological similarities to the Paeoniaceae subfamily and Dilleniidae subclass, and it is thus considered to be an ancestral feature in the Caryophyllales [27,67].

Generally, cacti ovaries develop multiple ovules [68], but *Pereskia aculeata* develops fewer than five ovules [27,49]. Ovule development in cacti exhibits similar characteristics to other members of the Caryophyllales, such as the hook-like shape of the carpels, which is also reported in *Phytolacca*, as well as the ovules’ primordium at the base of the ovary, such as that seen in *Phytolacca* and *Tetragonia* [50]. Another feature is the secondary augmentation of the ovules along the cross-zone, as in *Trianthema* and *Mesembryanthemum* [50]. The structure that has no homologs in other members of Caryophyllales is the pericarpel, a tissue that covers the ovary and seemingly is ontogenetically related to the stem. The ontogeny of this structure, together with the perianth, deserves further attention due to its likely complex evolution. In the next section, we will summarize the current understanding of perianth development and discuss its relationship with the pericarpel.

## 4. Sepals or Petals as Cacti Perianth Organs, or Neither?

While a double perianth with outer green sepals and inner colored petals is a well-established feature in the core eudicots [69], perianths with only one whorl or with more than two whorls also occur [1], a feature that has been documented in species within the Caryophyllales. In this order, such a simple classification can be deceiving, as some organs with equivalent functions can have different developmental origins, and sepals and petals cannot be distinguished on the basis of the presence or absence of pigmentation [68]. In addition, a number of families in the Caryophyllales are characterized by a simple perianth, but contrary to expectations, the whorl that is missing corresponds to the petals, not the sepals [70,71].

The discussion on the origin of the perianth across angiosperms and core eudicots, and particularly in Caryophyllales, has received a great deal of attention [17,69,72,73,74], and multiple independent origins of sepals and petals have been reported. Ronse De Craene [69] suggested that petals have been independently lost and “reinvented” at least five times in Stegnospermataceae, Aizoaceae, Portulacaceae, Caryophyllaceae and Molluginaceae. Similarly, Brockington et al. [37] suggested at least nine independent origins of a differentiated perianth within the Caryophyllales. An example of this phenomenon can be found in *Delosperma napiforme*, where the outer stamen primordia develop into sterile staminodes and become increasingly petaloid in a centrifugal pattern, resulting in many white and showy petal-like staminodes in the mature flower, which contrast with the green sepals [15].

Cactaceae represent a highly derived clade with an increased number of petaloid sepals [41], which are indeterminate and polymerous [68], developing in a centripetal order [50] with a spiral phyllotaxis [37]. Ronse De Craene [72] considered that the perianth in Cactaceae is formed by sepals and not by petals. Due to this, he denominated this structure as “petaloid sepals’’ and considered them nonhomologous to the petals present in species such as *A. thaliana* or *A. majus*. Thus, petaloidy of the sepal whorl is an important evolutionary phenomenon that has evolved either independently from modifications in the petal whorl or as a consequence of a reduction in the petal whorl [69]. As we mentioned above, the perianth in the cactus family has been considered to have a gradation from bracts to sepals to petals [23,50] or bracts to tepals [50,56]. This phenomenon has never been addressed in detail from a developmental or an ontogenetic perspective, but studying it would be instrumental to determine whether the perianth of a seemingly sepaloid origin present in this family is a product of a reduction in the petal whorl or if there is, in fact, a morphological gradation related to the differential expression of transcription factors directing perianth development that favors the interpretation of a transition from bracts to sepals and petals within a cactus flower (Figure 3). This last hypothesis, the loss of or reduction in petal primordia, would suggest different underlying morphogenetic scenarios and would also potentially shed light on the ontogeny of the flower shoot (Figure 4).

In many cacti, flower shoots exhibit a gradual development of flower-associated structures. For example, from the base of the flower shoot to its apex, some bracts as small as scales grow in size to the outer perianth segments. That is, these “outgrowths” go from the outside to the inside. While they are small, growing in the basal-most part of the pericarpel, they look like leaves, and they turn into sepal-like structures when they reach their maximum size, in the most distal part of the flower shoot [50]. Some examples of gradual modification of bracts into sepaloid or petaloid organs are seen in *Ferocactus*, *Selenicereus* or *Polaskia* (Figure 3). The “flower shoots” in these genera show green bract-like or scale structures in the outermost section of the perianth, gradually changing color and texture as subsequent whorls develop and ultimately developing a petal-like morphology in the innermost whorl. This gradation was interpreted by Buxbaum [50] as foliar organs becoming petaloid at the perianth section of the flower shoot. In contrast, in *Pereskia*, species such as *P. aculeata* have an abrupt transition from green bracts to petaloid perianth segments [27], akin to having a double perianth (with differentiated sepals and petals). This abrupt transition between green sepaloid and pigmented petaloid perianth organs is also observed in *L. ziniiflora* (Figure 1, green clade).

In cases where the vasculature has been analyzed, it has been documented that in *Opuntia*, tepals show multiple vascular bundles derived from a central bundle, which differs in size from the remaining bundles, with the central bundle being larger than the others. These vascular bundles resemble those seen in the leaves of some cacti and are arranged or located in a collateral manner. Such an arrangement can also be seen in vascular bundles of the cortical and medullary bundles in the stems of Cactoideae species [59]. In *P. aculeata* [27], bracts and tepals can be distinguished because the procambium distribution is different: the midvein is prominent in bracts, while in tepals, it is small. Vascular bundles have been used to distinguish the origin of petal-like structures in other species. Nevertheless, Ronse De Craene [41] argued that using the vasculature to distinguish any flower-associated laminar structure as either petal or sepal is far from accurate.

The lack of distinction between bract scales and sepaloid and petaloid structures described by several botanists, and therefore referred to as tepals, led us to consider whether the genetic mechanisms that control the identity of these organs can be used to better assign their type of floral whorl (Figure 4). It is likely that evolutionary modifications of the ABC model, which describes the mechanism of whorl identity in *A. thaliana* (described above), could be related to perianth morphology in the Cactaceae. In cacti, one possibility is that the orthologs of B class genes are not involved in the determination of tepals, therefore lacking petaloid identity, so tepals could, in fact, be modified sepals (Figure 4; Hypothesis I). In this scenario, class A genes could be the only ones acting in the determination of these organs, resulting in a reduction in or overall loss of petals in the perianth of cacti. This is the case for species in Ranunculales, where the loss of petals correlates with the decreased or overall lack of expression of the class B gene *AP3*-like [12,76]. A second possibility is that genes from classes A and B are expressed in an intergraded manner, consistent with the “fading borders” variation of the ABC model proposed for some species of *Nelumbo*, which also have a spiral arrangement of perianth organs [75,77]. In Cactaceae, it would only apply to the sterile perianth-like organs preceding the staminodial ring meristem (Figure 4; Hypothesis II).

Given that *Pereskia* and *Leuenbergeria* exhibit an abrupt transition from bracts to pigmented tepals not observed in other subfamilies, there is the possibility that the genetic control of perianth development in these genera differs from the mechanisms proposed above for clades such as Opuntioideae and Cactoideae. A third possibility is that components of the perianth are highly modified bracts, whose development would not involve the activity of class A or class B genes (Figure 4; Hypothesis III).

In this regard, comparative genetics studies highlight that in many angiosperms, the corolla does not need B or A class gene function to develop [78,79], while A class genes seem to exert a cadastral function that limits the expression of B and C class genes into the outer whorl(s) of many flower species rather than playing an important role in petal development, as was originally proposed based on the model species *A. thaliana* [78]. This could very well be the case in Cactaceae, as studies of the spatiotemporal expression of B class genes in developing flowers of closely related families show that their petaloid perianth develops by other means aside from B class gene function [15]. At the ordinal level, there are certain lines of evidence showing a diversity of genetic mechanisms that could determine the perianth in Caryophyllales. For example, Brockington et al. [15] analyzed B and C class gene expression in *Sesuvium portulacastrum* and *Delosperma napiforme* (Aizoaceae) using mRNA in situ hybridization of the orthologs of *AP3*, *PI* and *AG* to survey their role in determining petal identity. These authors reported no expression of *SpPI* and *SpAP3* in developing tepals; rather, these genes were expressed during the development of the androecium and gynoecium of *S. portulacastrum*. In addition, they found that in *D. napiforme*, a species with stamen-derived petals or andropetals, *DnPI* is expressed less intensely in stamen primordia that will give way to the innermost andropetals, while *DnAP3* is expressed more strongly in stamen primordia that will develop as outer andropetals. Furthermore, both *DnPI* and *DnAP 3* show only early and transient expression in andropetal primordia; thus, no heterotopic gene expression pattern of B class genes explains petaloid perianth development in these two species [15]. Hence, evidence of the homoplastic perianth origin across the Caryophyllales together with their experimental data led Brockington and colleagues [15] to propose that petaloid evolution in this order as well as in other angiosperms could occur by developmental mechanisms alternative to those proposed in the ABC model of floral development. In the case of cacti, gene expression studies of B and C class genes in the developing flower primordium could not only aid in analyzing whether B class genes play a role in the morphological gradient of petaloid organs observed on the outer surface of the pericarpel but also help to unravel the ontogeny of the “petal” primordia that can be observed developing on the outer rim of the ring primordium where stamens develop, as documented in some studies [57,80].

In this regard, it has been acknowledged by several authors that the perianth is the most plastic organ in flowering plants, with multiple independent examples of gains and losses, a phenomenon that is particularly acute in Caryophyllales [41,81]. Although petals are often topographically defined as the inner whorl in the perianth, this is clearly limited and unreliable, as it does not provide any information on homology [69]. In contrast, distinct evolutionary origins of a differentiated perianth, with contrasting petal derivations, do provide the necessary variation and evolutionary replicates to assess the role of variations in the canonical eudicot petal identity program in recurrent petal evolution [15].

## 5. The Flower Shoot

As mentioned above, the flower in Cactaceae is referred to as a flower shoot because of the apparent vegetative origin of the pericarpel. However, the ontogenetic origin of this tissue is still unclear. A hint towards comprehending this issue could lie in understanding the origin of the inferior ovary in Cactaceae. Two different hypotheses of the origin of the inferior ovary have been proposed. The first is the appendicular origin of the inferior ovary, which proposes that the nature of the external ovary wall originated from the fusion of the concrescence from the bases of the calyx, corolla, androecium and gynoecium. The second hypothesis is the receptacular origin of the inferior ovary, which posits that the external ovary wall has an axial-vegetative nature [25,82]. Both hypotheses have been discussed for different groups of plants, including Cactaceae, but only the second hypothesis might be able to explain the origin of the inferior ovary in this family [82]. Nevertheless, axial tissue covers the inferior ovary in Cactaceae, but stamens and tepals are also embedded in it [60,61]. Such morphological arrangement explains why the floral structure in cacti is often called a “flower shoot” [60].

This concept of the flower shoot or the assembly between vegetative and floral tissues has been widely accepted in cacti [60,61,83]; however, no formal study has been conducted to determine the developmental and evolutionary origins of the pericarpel. The term pericarpel was proposed by Buxbaum [50] to define the receptacle tissue that surrounds the carpels. The pericarpel in some genera (i.e., *Selenicereus* and *Epiphyllum*) is prominently prolonged, forming a tube, but it is considered nonhomologous to a flower tube because it is an extension of the pericarpel, which is considered to be of a vegetative origin [50,84]. Although several authors often use the term pericarpel, mainly in Cactoideae [18,23,56,62], Leuenberger [49] argued that this distinction is not feasible in *Pereskia*, where the androecium and gynoecium are not hidden inside the axial tissue, nor are the sterile perianth whorls. Consequently, he continues to call this the receptacle or receptacle cup.

To date, multiple pieces of evidence support the so-called flower shoot concept: the presence of laminar leaves on the pericarpel [61], which have the same developmental pattern as in vegetative branches observed in *Opuntia* [59] and *Pereskiopsis* (Figure 5a); the presence of vegetative organs such as bracts or spines on the pericarpel (i.e., *Echinocereus*, Figure 5b); and the elongation of the “floral tube” in many species (i.e., *Aporocactus*, Figure 5c), which supports the idea that the “floral organs” are surrounded by “axial tissue” [26].

The pericarpel can also bear areoles with woolly hairs (trichomes) and a few spines or bristles in their axils [23]. Furthermore, the areole is considered homologous to an axillary bud [50,85,86] but not simply an axillary bud. Areoles are unique to cacti [23] and emerge in the axils of developing leaves or tubercles, and they are distinctive because they can originate spines, glochids (deciduous spines) and masses of trichomes that can be short or have long hair-like structures. Areoles also have the capability to originate flowers, branches and roots [57,87]. It is believed that areoles are complex axillary meristems that are composed of collapsed shoot nodes and internodes; therefore, the areole is not a single axillary meristem but rather a group of several axillary meristems [60]. Hence, the areole might have different functional meristematic domains, each of which could originate from different organs. The term areole is useful because the bud’s spines persist even if the axillary bud meristem goes on to produce a flower and fruit [60]. Flowering in most angiosperms causes bud scale abscission; therefore, after the fruit is shed, the region is little more than a set of scars, but in cacti, the entire set of spines is still present [60]. Furthermore, some cacti produce spines for a prolonged period, longer than most axillary buds, which produce bud scales; therefore, these growing structures are more appropriately considered short shoots rather than merely buds [60]. Thus, the presence of areoles over the pericarpel indicates that this structure is more complex than just a receptacle.

The presence of podaria (an enlarged leaf pedicel), tubercles and ribs (fused enlarged leaf pedicels) in the flower shoot has been taken as additional evidence to support the axial origin of the pericarpel [50] because these structures are typical of stems in the Cactaceae family. For example, anatomical similarities were observed in the stem and the pericarpel of some *Opuntia* species and other members of the Opuntioideae subfamily, such as pericarpel epidermal tissue, hypodermis, cortex and vascular tissue [59,88]. While these observations are suggestive, we caution that mere morphological and anatomical observations might not provide robust evidence to argue for the developmental and evolutionary origins of the pericarpel.

In summary, the concept of flower shoots has been commonly used by most scholars to describe the singular reproductive units in Cactaceae; nevertheless, the ontogeny of this structure has been vastly understudied, leaving two key innovations to be further investigated in this unique plant family: first, the pericarpel proper, which we interpret as a unique structure developed from a process of synorganization between axial and reproductive structures (Figure 2); second, a spiral perianth where a continuous intergradation of bracts to sepaloid to petaloid structures takes place in an acropetal manner throughout the flower shoot, together forming a unique type of seemingly terminal flower in the majority of cacti species (Figure 4 and Figure 5). These phenomena are likely intertwined and could be the product of the blurring of different kinds of boundaries: those related to the differentiation of floral organ whorls (as contemplated in the ABC model discussed above), likely enabled by the existence of an androecial ring primordium, and the unique ontogeny of an inferior ovary that apparently becomes fused with the underlying vegetative axis in what has been construed as a terminal flower. We posit that the close proximity and spatial disposition of axillary meristems within an areole, which is in itself a compressed branch, could be a proximal explanation for the apparent synorganization of the reproductive axes in cacti species. In the following sections, we review the available developmental data for each of these structures and propose developmental hypotheses of how flower shoots could be formed. Furthermore, we discuss different experimental approaches that could help test these ontogenetic hypotheses and help to unravel the ontogeny of this unique reproductive unit.

## 6. The Unique Pericarpel of Cactaceae

One of the few authors to articulate a developmental explanation for the origin of the flower shoot in cacti is Mauseth [61], who explained how the cactus flower ended up being a flower shoot through a metaphor of an elongated party balloon, in which the floral whorls are positioned at different levels outside the balloon: the lower half bears leaves; the middle part bears sepals, and “petals’’ are placed above this section, leaving stamens and carpels to develop on the top of the balloon. In other words, this could be interpreted as an elongated flower primordium. Then, the author proposed that a force pushes all the elements of the flower inside the balloon (into the axis of the stem) until the entire flower ends up inside, externally covered by leaves. This proposal of the invagination of floral whorls into the floral tube of a cacti flower shoot is an interesting idea to explore; however, it should be noted that in some anatomical sections of *Echinocereus*, it appears that the floral meristem forms within the areole (vegetative meristem), giving way to carpels, stamens and petaloid organs while still below the areole surface, in the “inside” of the developing reproductive unit [35,57,80,89]. The invagination of the ovary, together with the differential growth of the style, takes place in later stages [80]. Further during development, the residing ovary seems to fuse with the surrounding stem tissue, forming the internal part of the pericarpel. Additional evidence of the integrated nature of this structure is the fact that the perianth vasculature arches beneath the carpel [80]; that is, the vascular supply to the placentae is derived from the recurrent receptacular system, which diverges downward after providing traces to the perianth and androecium and from which the dorsal and ventral carpellary bundles diverge at the level of the ovary roof [80]. While some authors suggest that a precondition for synorganization is the existence of floral organs organized in whorls [90], this particular case entails fusion with an organ (the carpel) that sits on top and becomes progressively embedded into the surrounding vegetative tissue (Figure 5).

Fusion between different structures has been considered an important macroevolutionary trend in angiosperms, with potential adaptive implications [91,92]. In the case of cacti, the protected nature of the inferior ovary, as well as other floral organs embedded and protected by the lateral organs that develop on the external part of the pericarpel (i.e., areoles, or bracts), has been considered an important adaptation to the extreme environments where many cacti species dwell [23,50,56]. The pericarpel thus entails the integration of axial and reproductive tissue, a phenomenon whose underlying genetic mechanisms could entail a blurring of boundaries between floral and axial meristems. This phenomenon could be related to the close proximity of several meristematic tissues within an areole: spine meristem, axial meristem and leaf meristem [50,89], as well as the identity of the areole, being either a vegetative meristem, an inflorescence meristem or a floral meristem, or a mixture of several types of meristems as described above.

During the reproductive transition, vegetative meristems receive cues for their developmental conversion into reproductive meristems. Some of these cues are sparked by so-called florigen genes, such as *CONSTANS* (*CO*), *FLOWERING LOCUS T* (*FT*) and *FLOWERING LOCUS D* (*FD*) [93]. When vegetative meristems become competent to produce reproductive structures, they turn into inflorescence meristems. In angiosperms, inflorescence meristems can have different degrees of vegetativeness [94], producing many floral meristems, as in indeterminate inflorescences (i.e., *Arabidopsis* and *Antirrhinum*), or a single floral meristem, as in solitary flowers (i.e., tulips or hibiscus). The degree of vegetativeness of an inflorescence seems to be controlled by a gene known in *Arabidopsis* as *TERMINAL FLOWER 1* or *TFL1* [14,95] and its ortholog in *Antirrhinum*, *CENTRORADIALIS* or *CEN* [96,97,98]. In both species, mutants in these genes cause premature exhaustion of the inflorescence meristem, giving way to a terminal flower. While TFL1/CEN maintain an undifferentiated inflorescence meristem, they antagonize the transcription factor *LEAFY* (*LFY*) in *Arabidopsis* [99] or *FLORICAULA* (*FLO*) in *Antirrhinum* [100], whose activity gives floral meristem identity to the inflorescence meristem, by activating floral whorl identity genes (mentioned in the ABC model). Loss of function in LFY/FLO results in plants possessing what seem to be inflorescences but are instead holding bracts or leaves instead of flowers. In other words, once the reproductive transition has occurred and the vegetative meristem transitions into an inflorescence meristem, *TLF1*/*CEN* maintain meristem vegetativeness, while *LFY/FLO* promote meristem determinacy towards a floral meristem. Although the activities of these genes have been primarily studied in model species, their function seems to be conserved across many angiosperms, and therefore these genes have often been used to determine, for example, whether some structures are flowers or inflorescences [101].

To ascertain what type of meristematic tissue is present in the pericarpel (and associated tissues in the mature flower shoot), we propose and discuss three alternative scenarios. Our first hypothesis is the simplest one, where the induction of a floral meristem within an areole could induce the growth of a nearby axial meristem that eventually fuses onto the basal part of the developing floral meristem, giving way to a compressed branch of nonreproductive origin, which would be the pericarpel (Figure 6a). This proposal surmises that the pericarpel has a purely vegetative origin, while the floral whorls are on top and embedded. In this scenario, *LFY/FLO* expression would be expected in the tissue that originates from the floral whorls but not in the pericarpelar tissue surrounding the ovary. *TFL1/CEN* expression would also be expected to transiently precede *LFY/FLO* expression but disappear afterwards as the inflorescence meristem is consumed. This also implies that the pericarpel has a vegetative origin and that gene activities in that tissue would resemble the stem. In the second scenario, the inflorescence meristem (IM) differentiates within the areole niche and protrudes to form the flower shoot, differentiating into a floral meristem in the distal part of the protruding tissue, while it remains somewhat undifferentiated in the proximal part of the meristem, where it develops into a highly modified inflorescence with a terminal flower, where a series of bract-like structures develop on the outside, and parenchymatous and vascular tissues develop in the inside, forming the pericarpel (Figure 6b). This would mean that the flower shoot is, in fact, the outcome of the inflorescence meristem, with a terminal flower and several axillary vegetative buds (areoles) constituting the pericarpel. Similar to the previous scenario, orthologs of LFY/FLO would be expected on the tissues that will originate the floral whorls; however, orthologs of TFL1/CEN would be expected to be active on the pericarpel tissue, maintaining the vegetativeness of axillary meristems (areoles). However, this hypothesis does not explain how the axillary buds remain dormant; therefore, they do not produce inflorescence branches, which might then be controlled by a different mechanism. In the third scenario, the entire flower shoot could originate directly from a floral meristem (FM), with a transient inflorescence meristem state, and thus the ontogenetic origin of the pericarpel would be floral (Figure 6c). This would mean that orthologs of *LFY/FLO* are active in the tissue that conforms to the pericarpel as well as the tissue that will originate from the floral whorls. This hypothesis implies that bracts on the pericarpel, as in *Ferocactus* (Figure 3b), have a floral whorl identity.

While, in advanced stages of development, the branch-like nature of the pericarpel manifests through the formation of areoles (Figure 5) on its surface, many cacti species show a morphological gradient of bracteoid–sepaloid–petaloid organs towards the distal part of the flower shoot, commonly organized into a structure that resembles a perianth (Figure 4). It is possible that concomitant with the internal synorganization taking place between the vascular tissue of the inferior ovary and the internal part of the pericarpel, on the outside, a basipetal hormonal and transcriptional gradient can induce the progressive transformation of bracts into petaloid lamina.

Another phenomenon that warrants attention and that could enable us to ascertain if the developing flower primordium becomes embedded into the pericarpel tissue or if the pericarpel tissue grows around the developing flower is analyzing the very early stages of flower development in several species. Nevertheless, this is not straightforward, as the cacti flower meristem is often difficult to recognize on the surface or even develops while initially hidden within the areole, as is the case of *Echinocereus* species, where the areole with the flower meristem is engulfed by the stem tissue [89]. Thus, two possible scenarios are proposed: in the first scenario, the flower meristem forms a bulge that protrudes from the surface of the stem, as is common in most flower meristems (Figure 7a), but the differential growth rate of the underlying surrounding tissue and the process of synorganization translates into a floral meristem that is embedded within the developing pericarpelar tissue. An alternative scenario would be one where the flower meristem invaginates into the areole and fuses with the surrounding meristem (light green, Figure 7b), then both meristem types (flower meristem and surrounding vegetative meristem) grow together and the floral meristem is thrust outwards by the surrounding tissue, which has a higher growth rate, and eventually covers the developing flower. Both proposals have in common that we assume that the pericarpel tissue is axial in origin and that a process of synorganization takes place early on, with the surrounding tissue growing faster than the floral tissue, but in the first case, the floral meristem is covered by the axial tissue, while in the second case, it initially appears to be embedded into it.

At the core of our proposal is the notion of a particular case of synorganization between floral and axial tissue, where “floral identity” genes are likely necessary for reproductive organ formation but not for perianth development; thus, A class genes would be involved in boundary delimitation [85]. In this context, genes involved in organ fusion/boundary delimitation, such as the orthologs of *CUP-SHAPED COTYLEDON* and *NO APICAL MERISTEM* (*NAM*), which are part of an NAC-domain family of transcription factors that are present in all angiosperms analyzed to date, could be playing a role in floral fusion. *A. thaliana* mutants of *CUC* and *NAM* have shown partial to complete cotyledon fusion and reduced to complete meristematic inactivity [102]. Therefore, these genes have been shown to be involved in embryonic development, floral organ boundary specification [103,104], carpel development [105] and meristem delimitation [106]. Hence, they could be interesting candidates to analyze in Cactaceae and allied families. The spatiotemporal analysis of gene expression as well as the epigenetic regulation of orthologs of *CUC* and *NAM* genes [106] in all meristems present in an areole, as well as in developing flower shoots across select cacti species, could shed light on the ontogenetic mechanisms that underlie the formation of this unique reproductive unit, where synorganization seems to be taking place.

Furthermore, the interplay between these genes and others that are involved in maintaining the identity of the shoot apical meristem *SHOOT-MERISTEMLESS* (*STM*) [107], the transition from an inflorescence meristem to a single terminal flower meristem (*TERMINAL FLOWER1* (*TFL1*); [95]) and their interaction with the plant-specific *LEAFY* (*LFY*) gene, which is fundamental for flower meristem determination and organ boundary formation in *Arabidopsis* [99], through interaction with ABC class genes [14], would also be informative, as it is pivotal in floral development [14]. In addition, *LFY* has diverse roles in different angiosperms [108] and has been proposed to be a useful marker for inflorescence/flower boundary determination in other angiosperms, where it appears to play a central role in the initiation of angiosperm flowers, although other factors can be responsible for detailed floral patterning [109]. In the case of Cactaceae, the analysis of gene expression and protein interactions of the LFY ortholog could also help shed light on the morphogenetic identity of the seemingly compressed flower shoot. However, LFY expression must be considered with caution, as its expression does not always correlate with floral meristems [110].

## 7. Perspectives

The conspicuous characteristics observed in flower shoots of cacti, such as the presence of multiple perianth series with a possible sepaloid/bracteoid origin and the existence of a unique structure termed the pericarpel, make cacti flower shoots an exciting model to further our insights into the different developmental venues that underlie flower diversification in angiosperms.

The ABC model proposed a generic mechanism for flower determination that has been a useful conceptual framework to test for the genetic basis of homologous organs in many angiosperms. Recent studies in a diverse set of flowering plants have suggested that several variations of the *Arabidopsis*/*Antirrhinum* model exist, particularly with respect to the perianth whorls [111,112,113,114]. In Cactaceae, the apparent lack of homology of perianth structures with respect to *Arabidopsis*, as well as the unique pericarpel structure, suggests that cacti could represent an additional variation on a theme involving synorganization between reproductive and vegetative tissues; this must be further investigated through comparative gene expression and developmental genetic studies that could help ascertain the conserved functions and unique variations present in a structure whose morphological and histological identities need further study. In this regard, the pericarpel deserves further attention within its phylogenetic context, as we do not know whether it has homologous structures in Caryophyllales or in other angiosperm families. Different techniques can help us test the four main sets of hypotheses presented here, namely, the ontogenetic identity of the perianth (Figure 5), the identity of the pericarpel (Figure 6), how/if the flower became embedded (Figure 7) and the mode of synorganization between axial and flower structures. In situ hybridizations of orthologs of ABC class genes in different stages of reproductive unit development, as well as *CUC*, *STM*, *LFY* and *TFL1*, could yield information regarding their spatiotemporal expression and possible interactions; yeast one- and two-hybrid experiments could provide information pertaining to protein–protein interactions, while transcriptome analyses of different tissues, such as bracts, leaves, spines, tepals, the ring primordium, the carpel and the pericarpel, could help unravel the genetic mechanisms important in cactus flower development. Additionally, complementation studies using candidate genes in cacti expressed in plant model species (i.e., *Arabidopsis* or *Solanum*) could shed light on the functional diversification of candidate genes. Mutagenesis in cacti has not been implemented; however, some species with aberrant flowers can be found, sometimes in natural populations [48] or in artificially obtained hybrids. This is the case for *Astrophytum* or *Epiphyllum* hybrids, where the perianth displays a wide range of phenotypes, adding functional data to shed light on the molecular basis of the perianth and the pericarpel.

Despite the popularity of cacti as ornamentals due to their charismatic features, as well as the economic value of products from the *Opuntia* genus, knowledge of their preanthetic developmental patterns is still incipient. Although considerable efforts have been made to resolve the phylogenetic relationships between different genera and species, with the objective of understanding the evolution of these spectacular plants, these efforts are still far from providing a full comprehension of their floral development.

## Figures and Tables

**Figure 1 plants-10-01134-f001:**
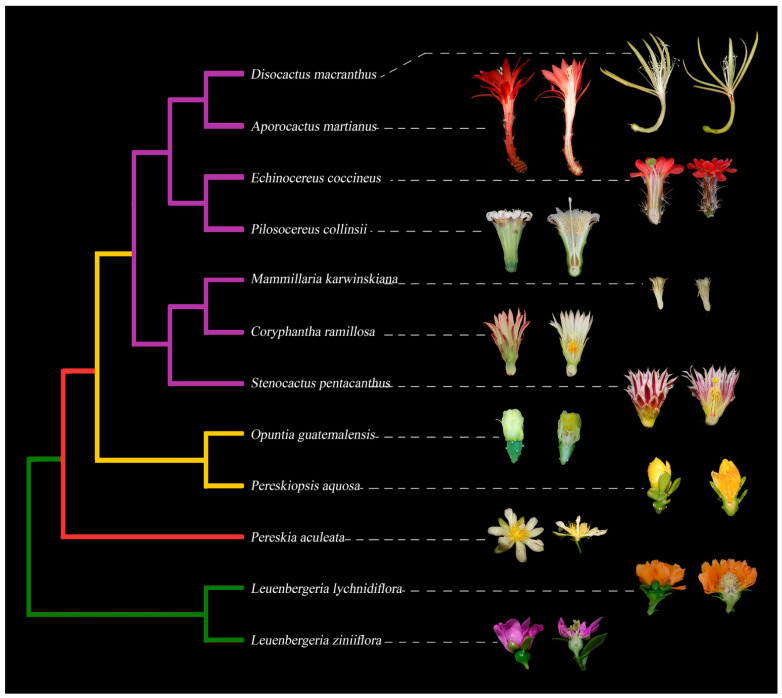
Simplified phylogeny of Cactaceae with representative flowers in longitudinal sections, exemplifying the diversity in floral morphology across the family. Leuenbergerioideae (green clade): *Leuenbergeria zinniiflora*, flowers with a hypanthium-like structure covering the ovary, with green sepal-like whorls and purple petal-like whorls; *L. lychnydiflora*, with green sepal-like whorls and orange petal-like whorls. Pereskioideae (red clade): *P. aculeata*, flowers with receptacle with areoles covering the superior ovary. Opuntioideae (yellow clade): *Pereskiopsis aquosa*, flowers with areoles and laminar leaves over the pericarpel; *Opuntia guatemalensis*, flowers with a succulent and thick pericarpel, foliar organs becoming tepaloid towards the apex. Cactoideae (purple clade): *S. pentacanthus*, *C. ramillosa* and *M. karwinskiana* showing a very reduced campanulate receptacle and a reduced pericarpel without areoles, spines and bracts; *P. collinsii*, showing a campanulate receptacle, without spines in the pericarpel and decurrent podaries; *E. coccineus*, showing a campanulate receptacle with a green spiny pericarpel and a red perianth; *A. martianus* showing a long tubular receptacle with bracts, spines and a red pericarpel, as well as a red perianth; *D. macranthus*, with a campanulate receptacle and a long tube, with a very reduced pericarpel and areoles, with a yellow perianth.

**Figure 2 plants-10-01134-f002:**
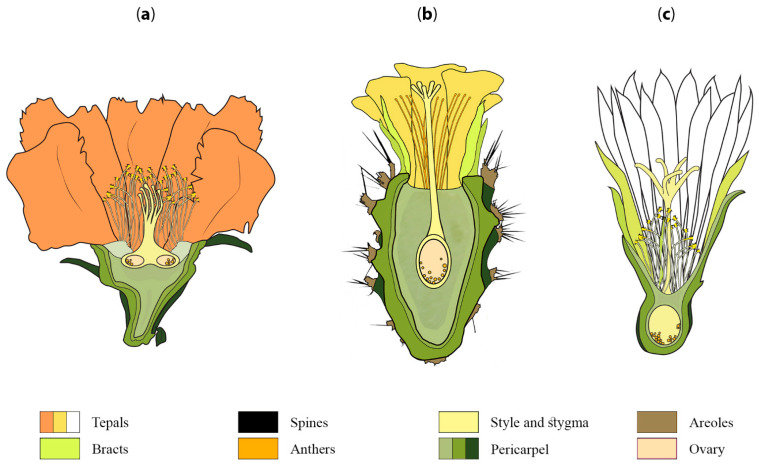
Schematic representation of three different flowers of Cactaceae. Longitudinal section through a flower of (**a**) *Leuenbergeria lychnidiflora* (Leuenbergerioideae), (**b**) *Opuntia guatemalensis* (Opuntioideae) and (**c**) *Coryphantha delicata* (Cactoideae) illustrating the variation in morphology found in cactus flowers.

**Figure 3 plants-10-01134-f003:**
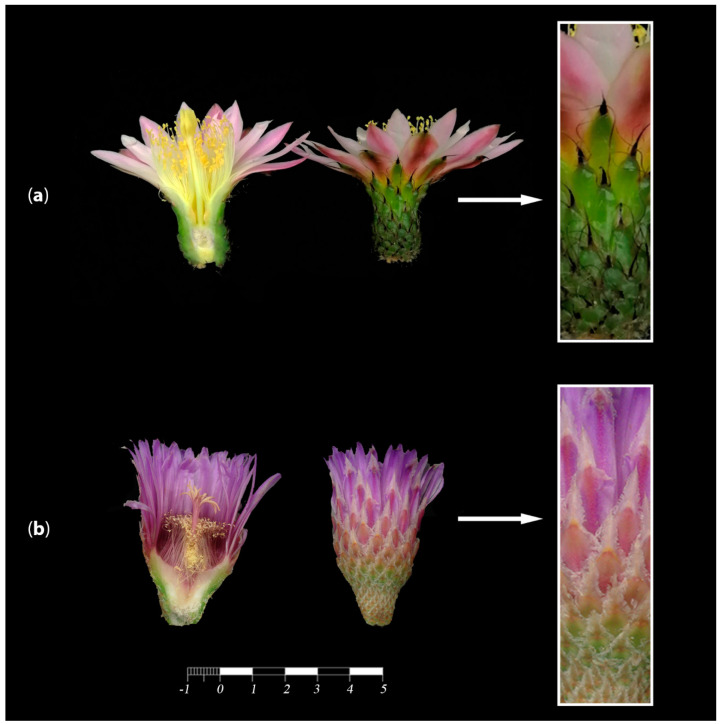
Development of bracteoid and perianth-like structures from the base to the apex of the flower shoot. (**a**) *Polaskia chende* with sepal-like structures that cover the branch flower, which abruptly become petal-like elements of the perianth. (**b**) *Ferocactus latispinus* with papery bracts at the base of the flower shoot that gradually become elements of the perianth (better referred to as perigonium, a term used when there is no clear differentiation between sepals and petals).

**Figure 4 plants-10-01134-f004:**
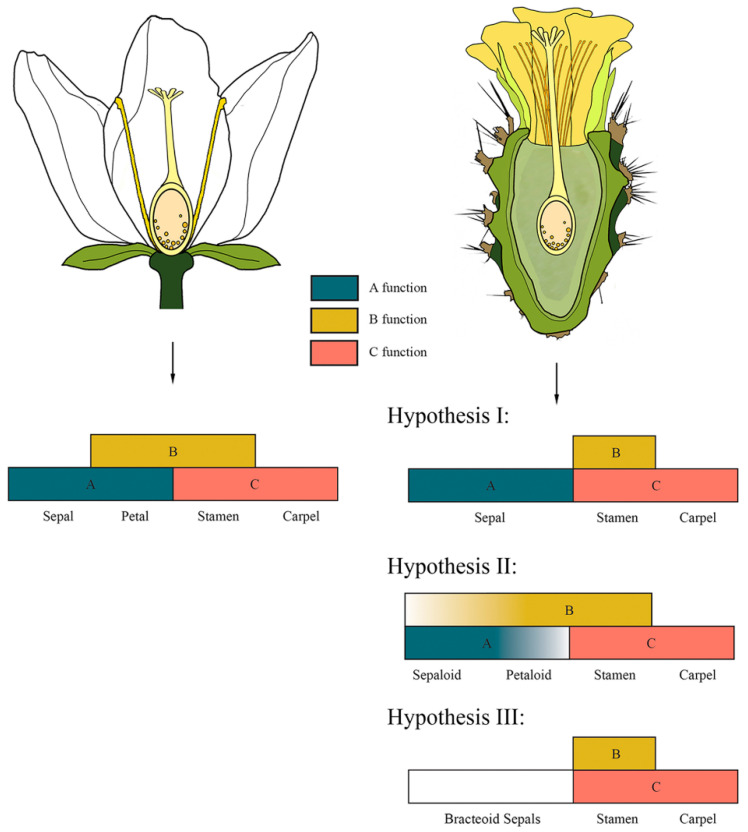
Canonical ABC model of organ identity vs. hypothesis of possible variations of the ABC model in a cactus flower. As the genetic identity of sterile structures in Cactaceae is not well resolved, we propose two hypotheses to explain the origin of these organs. Hypothesis I: Only genes from class A determine the perianth segments; thus, laminar sterile structures are modified sepals, while genes of class B are only expressed in stamen primordia (shifting boundaries model). Hypothesis II: As several authors have described the perianth segments in Cactaceae as intergraded structures that go from bracts to sepals to (petaloid) tepals [75] and are denominated sepaloid petals, we propose that the expression of class A and B genes is consistent with the “fading borders model” but applied only to sterile perianth organs. Hypothesis III: There is no expression of A-class genes and as such, the perianth is bracteoid.

**Figure 5 plants-10-01134-f005:**
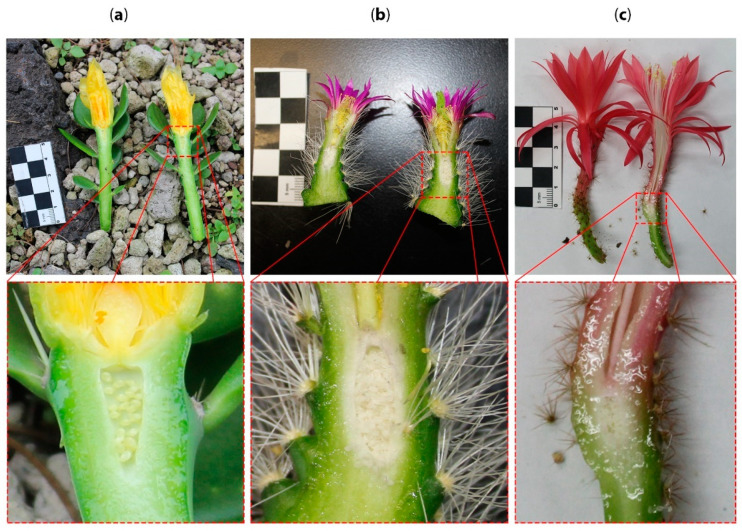
Flowers from different genera across Cactaceae where continuity between the stem and floral elements is evident. (**a**) *Pereskiopsis aquosa* (Opuntioideae, Cylindroputieae), stem looks like a branch with leaves, nodes and internodes topped by a flower. The ovary is sunken into the stem. In these examples, the flowers and the stem do not resemble distinct organs; rather, they display an intergradation of vegetative and flower-associated structures, with a sunken ovary. (**b**) *Echino-cereus parkerii* (Cactoideae, Pachycereeae), the stem narrows towards the apex where the flower develops. (**c**) *Aporocactus martianus* (Cactoideae, Hyloceareeae), the apex of the long stem is topped off with a red flower, and the ovary is sunken into the axial tissue.

**Figure 6 plants-10-01134-f006:**
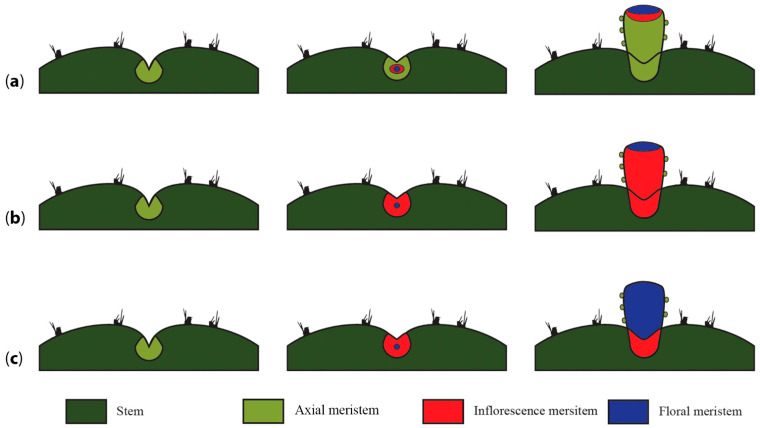
Alternative hypotheses of flower shoot inception and the origin of the pericarpel. (**a**) An axial meristem is induced to differentiate into an inflorescence meristem; this phenomenon induces a nearby vegetative meristem present in the same areole to grow and elongate around the basis and beneath the developing flower, eventually enclosing it and forming the vegetative part of the pedicel. (**b**) The inflorescence meristem gives way to the pericarpel, which will develop floral organs in the distal part of the flower branch and compacted branches (areoles) towards the proximal section where the reproductive unit inserts into the stem. (**c**) Once a meristem is induced to produce a flower, the inflorescence meristem gives way to a floral meristem that elongates with a peduncle that bears areoles instead of bracts.

**Figure 7 plants-10-01134-f007:**
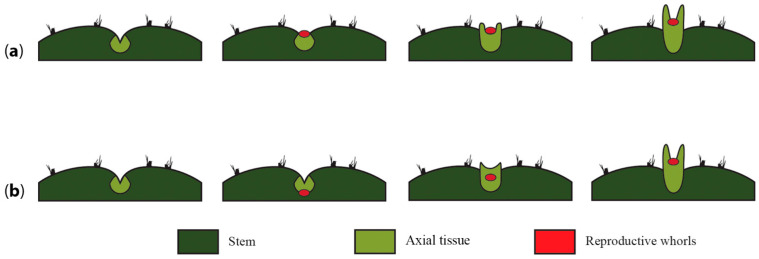
A flower sunken or covered by a shoot. (**a**) Once the axial meristem is induced to differentiate into a reproductive unit, the reproductive whorls (red dot) develop as a bulge protruding from the surface of the areole and induce the differentiation of lateral pericarpel tissue into a “branch” (light green structure). These two structures continue to develop, but accelerated growth of the axial tissue ends up surrounding and partially covering the developing floral meristem. (**b**) The reproductive whorls (red dot) invaginate into the areole and induce the development of lateral pericarpel tissue into a branch that eventually thrusts the inflorescence meristem outward, generating the flower shoot. Accelerated growth of the axial tissue ends up surrounding and partially covering the developing floral meristem.

## Data Availability

Not applicable.

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
