# Peer review of "Blurring the Boundaries between a Branch and a Flower: Potential Developmental Venues in CACTACEAE"

_plants, 2021, doi:10.3390/plants10061134_

Round 1
Reviewer 1 Report
Dear Editor
It is an important original paper but too tedious and suffered from extra citations.
Major reservations:
1. The paper needs a serious reduction, especially when the text is not related directly to the Cactaceae e.g. lines 263-300.
2. Section 7 is too lengthy and needs to be shortened at least by 30%.
3. Lines 595-625 seem too speculative and are not related directly to the Cactaceae. The authors are requested to sharpen this point.
4. The hypotheses are scattered in the text, maybe there is room to present all of them in one section at the end of the paper?
5. The research goals are not well explained and are too general.
Minor comments:
1. Scientific names have to be in italics!.
2. In several cases the references are not presented as numbers but with full names of the authors e.g. lines:117-118; 188; 241, 672
3. Line 422: Who is the first to raise this hypothesis.
4. Lines 472 and 486: Any supporting evidence or data to validate the
raised arguments.
5. Line 226: Why not cite Endreess directly, it is a modern available reference.
In general, it is an interesting paper although somewhat too speculative.
Suggest tightening the text and crystalizing the various hypotheses into one condensed section.
Reviewer 2 Report
Please consult the attached file for several comments to improve the manuscript (e.g. italicize names, sentence fragment, incorrect or misleading or vague statements). Additionally, I was surprised no mention was made of Calymmanthium, which has a striking flower development where the flower appears to burst out of a stem-like shoot. And no comparison was made between the core Cactoideae where flowers usually arise directly from stem areoles and the Cacteae where flowers often arise from the tubercle base. Finally, I am no expert on the genetics or ontogeny of flowers, so please be sure those parts are carefully evaluated.
